# Interaction of miR-155 with Human Serum Albumin: An Atomic Force Spectroscopy, Fluorescence, FRET, and Computational Modelling Evidence

**DOI:** 10.3390/ijms231810728

**Published:** 2022-09-14

**Authors:** Valentina Botti, Salvatore Cannistraro, Anna Rita Bizzarri

**Affiliations:** Biophysics and Nanoscience Centre, DEB, Università della Tuscia, Largo dell’Università, 01100 Viterbo, Italy

**Keywords:** human serum albumin, miR-155, AFS, fluorescence quenching, FRET, computational docking

## Abstract

This study investigated the interaction between Human Serum Albumin (HSA) and microRNA 155 (miR-155) through spectroscopic, nanoscopic and computational methods. Atomic force spectroscopy together with static and time-resolved fluorescence demonstrated the formation of an HSA/miR-155 complex characterized by a moderate affinity constant (K_A_ in the order of 10^4^ M^−1^). Förster Resonance Energy Transfer (FRET) experiments allowed us to measure a distance of (3.9 ± 0.2) nm between the lone HSA Trp214 and an acceptor dye bound to miR-155 within such a complex. This structural parameter, combined with computational docking and binding free energy calculations, led us to identify two possible models for the structure of the complex, both characterized by a topography in which miR-155 is located within two positively charged pockets of HSA. These results align with the interaction found for HSA and miR-4749, reinforcing the thesis that native HSA is a suitable miRNA carrier under physiological conditions for delivering to appropriate targets.

## 1. Introduction

MicroRNAs (miRNAs) are short, linear, single-stranded, noncoding oligonucleotides that can regulate the expression of genes controlling fundamental cell life processes, such as cell growth and proliferation, differentiation, stress response and apoptosis [1,2,3,4,5,6]. MiRNAs play their key role at post-transcriptional level mainly by interacting with the 3′ (UTR) of target messenger RNAs. At present, more than 2600 mature miRNAs have been identified [7]; however, a complete knowledge of their operating mechanisms is still far from being reached. Each miRNA has specific stable levels of expression depending on tissues and on developmental stage and, when they are dysregulated or abnormally expressed, the development of various pathologies, including cancers, can be induced [8,9]. Indeed, in cancer cells, miRNAs are often heavily overexpressed, and they may also act as oncogenes [9]. For these reasons, miRNAs have emerged as promising diagnostic and prognostic biomarkers as well as therapeutic targets [10]. Besides the defined canonical paths, miRNAs can operate through other mechanisms, whose investigation is still under way, deserving upcoming interest for applications in therapeutics. Along the direction of non-canonical mechanisms, it has been found that some miRNAs (miR-4749, miR21 and miR23a) directly interact with the tumor suppressor p53 [11,12,13]. Such a finding opens new perspectives in the understanding of the control mechanisms underlying the p53 role and can also inspire the development of new strategies for therapies based on the preservation and enhancement of the p53 tumor suppressor function. 

Human serum albumin (HSA) is the most abundant globular protein in plasma [14]. Indeed, HSA is a ubiquitous protein present at various concentrations in different body compartments and accounting for the largest part of the osmotic pressure of blood [15,16]. HSA plays a variety of physiological functions, among which are the provision of most extracellular antioxidant activity, regulation of intracellular and plasma pH and inhibition of pro-inflammatory pathways [17]. HSA is known to have an extremely high capability to non-covalently bind different ligands, (e.g., long-chain fatty acids, hormones, nutrients, metal ions, bile acids and nitric acid) and deliver them to specific targets [17]. On the other hand, the intrinsic properties of HSA, combined with its biocompatibility and easy nano-engineering, make it extremely suitable for applicative aims, such as delivery of systemic diagnostic and therapeutic compounds [18]. Furthermore, HSA binding significantly influences pharmacokinetics, as it can increase solubilization and circulation of hydrophobic drugs, as well as enhance their half-life in biologic fluids [19,20]. 

The possible interaction between native, unmodified HSA and miRNAs, largely present in the cell environment, deserves particular interest from different points of view: (i) upon binding miRNAs, HSA could play a regulatory role, by affecting the effective amount of miRNA available for detection. For instance, the expression of miR-146a/b in human renal cell carcinoma is upregulated by HSA [21]. (ii) HSA charged with miRNA could also play the role of carrier for miRNA-based therapeutics delivering for the treatment of different diseases, such as cancers, neurodegenerative disorders and also for tissue regeneration. (iii) The possibility that HSA could interfere with miRNAs should be evaluated for ultrasensitive detection of miRNAs in vivo.

In this connection, we have investigated the interaction of HSA with miR-155, a multifunctional miRNA-regulating B cell differentiation, development stages, etc. [22]. miRNA155 is overexpressed or mutated in various malignant tumor cells, such as hepatocellular carcinoma and breast and colon cancer, and it has been demonstrated that it plays a key role in the mammalian immune system [23]. On such a basis, miR-155 is a suitable biomarker for many types of cancers. In the present study, we have combined different biophysical techniques and methods according to the investigation procedures previously adopted to elicit the interaction of miR-4749 with HSA [24]. First, the formation of a specific complex between HSA and miR-155 has been assessed by applying Atomic Force Spectroscopy (AFS). AFS is a nanotechnological label-free technique that allows for the investigation of, under near physiological conditions and at a single molecule level, the unbinding kinetics of specific biocomplexes by exploiting its ability to sense very small molecular forces involving partner biorecognition [25,26]. The interaction properties between HSA and miR-155 have then been confirmed in bulk, by following the static and time-resolved fluorescence of the lone intrinsic tryptophan (Trp214) of HSA. Additionally, time-resolved Förster Resonance Energy Transfer (FRET) experiments allowed us to measure a distance of (3.9 ± 0.2) nm between Trp214 of HSA (acting as a donor) and the chromophore Atto390 (acceptor) bound to miR-155. Such structural information has been exploited to single out a possible structure for the HSA/miR-155 complex in the framework of computational modelling and free energy calculations. 

## 2. Results and Discussion

### 2.1. AFS Results

The formation of the HSA/miR-155 complex has been investigated by AFS through the analysis of the interaction properties between individual partner molecules, with both the partners anchored to a solid surface and by following the procedure reported in refs. [11,24]. Briefly, force curves have been acquired by approaching the cantilever, whose tip was functionalized with miR-155, towards the HSA-functionalized substrate. During such a phase, the molecules may undergo a biorecognition process, which can eventually lead to the formation of a specific complex. The approaching phase is stopped when a preset maximum force value is reached (here 0.7 nN), and then the cantilever is retracted from the substrate. When the cantilever spring force overcomes the interacting force, the cantilever jumps off, leading to the unbinding of the eventually formed complex, with the corresponding unbinding force being determined from the jump extension. 

Force curves, collected at five different loading rates (R), given by the product of the cantilever retraction velocity (v) and the spring constant of the entire system (k_syst_), have been first selected as being consistent with the unbinding of specific events, as described in Section 3.2. For each loading rate, the unbinding forces have been cast into a histogram. All the histograms exhibit a single peak curve whose maximum falls at unbinding forces below 100 N; an example related to the loading rate of 4.6 nN/s is shown in Figure 1A. The specificity of the observed unbinding events has been verified by performing a blocking experiment in which the force curves have been collected at the same loading rate, using a miR-155 functionalized tip against an HSA-functionalized substrate which had been previously incubated with miR-155; the corresponding histogram is shown in the inset of Figure 1A (grey columns). The ratio of the number of events related to specific unbinding events over the total recorded event is reduced from about 45% to 20% upon blocking. Furthermore, the shape of the histogram upon blocking does not significantly change; thus suggesting that the residual interaction is of the same nature as that recorded before blocking. These results give substantial support to the specificity of the interaction between HSA and miR-155.

To extract quantitative information on the HSA/miR-155 complex formation, we have determined the most probable unbinding force (F*) at each loading rate by fitting the respective unbinding force histograms with a Gaussian function. In fact, even if the expected analytical form of the unbinding force distribution slightly deviates from a Gaussian curve (see ref. [27]), a fit by a Gaussian is widely used in AFS data analysis since its peak provides a reliable value for the most probable unbinding force [25]. An example of the fit with a Gaussian function is shown in Figure 1A (black line). The most probable unbinding forces at the different loading rates have then been analyzed in the framework of the Bell and Evans model, according to which F* follows a linear trend as a function of the natural logarithm of the loading rate, as described by the Equation [28,29]: (1)$$F^{*}=\frac{k_{B}\;T}{x_{\beta }}\ln\frac{{r\;x}_{\beta }}{k_{\mathrm{off}}{\;k}_{B}\;T}$$
where k_B_ is the Boltzmann constant, T is the absolute temperature, k_off_ is the dissociation rate constant, and x_β_ is the width of the energy barrier along the direction of the applied force. 

Figure 1B shows the most probable unbinding forces, plotted as a function of the natural logarithm of the loading rate. We note a linear trend, as predicted by the Bell–Evans model. A fit of these data by Equation (1) has led us to determine k_off_ = (0.44 ± 0.05) s^−1^ and x_β_ = (0.73 ± 0.08) nm. First, we note that the occurrence of a single linear trend indicates a single energy barrier for the interaction process. The value found for k_off_, which is related to the lifetime, τ, of the complex (τ = 1/k_off_) reflects a dissociation process generally faster with respect to those found for specific biological complexes, while it is rather similar to those of transient complexes [30,31]. Such a result is consistent with the possibility that, upon binding to HSA, miR-155 could be easily released. At the same time, we note that x_β_ falls in the range usually found for biological complexes [25].

These data can also be compared with those found for the HSA/miR-4749 complex obtained by the same approach (k_off_ = (3.0 ± 0.7)∙10^−1^ s^−1^, and x_β_ =(0.26 ± 0.08) nm) [24]. The k_off_ value is practically the same in the two systems, while the energy barrier is higher for HSA/miR-155 with respect to HSA/miR-4749. These results suggest that, although the lifetime of complex formation between HSA and miR-4749 or miR-155 is almost the same, the details of the interaction are somewhat different. To estimate the affinity constant of the HSA/miR-155 complex, we have then evaluated the corresponding association rate constant (k_on_) by following the procedure given in ref. [32]. We found a k_on_ of ~10^4^ M^−1^ s^−1^, which is almost the same of that of other biomolecular systems [33]. On such a basis, we have finally determined the affinity constants K_A_ = k_on_/k_off1_ = 2.3·10^4^ M^−1^ for the HSA/miR-155 complex, again in agreement with the value obtained for the interaction between HSA and miR-4749 [24]. 

### 2.2. Static and Time-Resolved Fluorescence Results

Moving from the single-molecule to bulk regime, with interaction partners no longer anchored on support surfaces but free to diffuse in solution, the average tendency of miR-155 and HSA molecules to form an HSA/miR-155 complex has been assessed by fluorescence spectroscopy. Indeed, the lone tryptophan residue of HSA, Trp214, constitutes a solvent-exposed fluorophore [34], and a ground-state complex formation can be identified by the observation of a decrease in its fluorescence emission intensity accompanied by an unchanged fluorescence lifetime in the presence of a candidate binder (static quenching) [35]. Solutions of 5 µM HSA in Tris-HCl buffer pH = 7 with increasing miR-155 concentrations (0–50 µM) have been prepared as described in Section 3.3. Their steady-state emission spectra have been acquired by exciting Trp214 at 295 nm and revealing its fluorescence profile from 310 to 500 nm. HSA exhibits a maximum emission intensity at 346 nm (Figure 2A, black line), consistently with a full exposition of Trp214 to the solvent [34]. A noticeable decrease in the fluorescence has been observed upon progressively higher additions of miR-155 (colored lines in Figure 2A), with no significant wavelength shift of the emission peak; such a behavior indicates that the addition of miR-155 does not directly affect the region around Trp214. The plot of the relative fluorescence intensities, F_0_/F_i_, detected at 346 nm as a function of miR-155 concentration follows a linear trend (Figure 2B), according to the Stern–Volmer model. A fit of the experimental data by Equation (2) (reported in Section 3.3.1) has allowed us to extract a Stern–Volmer constant, K_SV_ of (2.6 ± 0.2)∙10^4^ M^−1^. 

To determine whether such a quenching can be attributable to a formation of an miR-155/HSA complex or due to collisions (dynamic quenching), the same samples have been analyzed by time-resolved fluorescence spectroscopy. In particular, fluorescence emission decays have been recorded by using an excitation wavelength of 295 nm. The acquired decays for HSA alone (black line) and of HSA in the presence of miR-155 with a concentration ratio of 1:1 (red line), are representatively shown in Figure 3. At visual inspection, the time-resolved emission curves of HSA in the presence of miR-155 show a slightly faster decay in comparison to that of HSA. The differences between the spectra emerge more clearly when these decays are carefully fitted by Equations (3) and (4) (see Section 3.3.2). The fitting has allowed us to obtain a very good description of the data, as evident from the fitting curves (see black lines in Figure 3, and from the weighted residuals, shown at the bottom of Figure 3). The goodness of the fit has been evaluated in terms of the χ^2^ value, which has been found to always be in the 1–1.1 range. The calculated average fluorescence lifetime values were: <τ_q_> = (5.38 ± 0.01)∙10^−9^ s for HSA alone and <τ_D_> = (5.33 ± 0.01)∙10^−9^ s for HSA/miR-155. Since a deviation of less than 4% has been observed, the value <τ_q_> of HSA is substantially maintained for HSA in the presence of miR-155 <τ_D_>; such a finding constitutes evidence of static quenching. As further confirmation of this, the bimolecular quenching constant, k_q_, derived from K_SV_/τ_q_, amounts to (4.43 ± 0.02)∙10^12^ M^−1^ s^−1^, two orders of magnitude higher than the typical diffusion-controlled quenching rate [35]. Therefore, the extracted K_SV_ effectively represents the affinity constant, K_A_, associated with the complex formation. 

In summary, for HSA and miR-155, fluorescence experiments support the establishment of an interaction under equilibrium conditions described by a constant compatible with that determined by AFS and similar to that observed for HSA and miR-4749 [24]. In addition, the fluorescence spectra suggest information concerning the interaction mode. Indeed, the absence of a shift in the emission peak at 346 nm throughout the titration (Figure 2) indicates that the binding of miR-155 does not affect the folding of HSA around the Trp214 residue. Such a behavior, similar to that also observed for the interaction between HSA and miR-4749 and between DBD-p53 and miR21 [11], can be described in terms of an allosteric mechanism. In particular, it can be hypothesized that the binding of the microRNA induces a conformational change of HSA without altering the Trp214 exposition to the solvent, with this affecting the emission fluorescence intensity as a result of a different coupling of the intrinsic chromophore with the biomolecule skeleton.

### 2.3. FRET Results

To gather further and more in-depth insights about the structure of the HSA/miR-155 complex, we have turned to FRET experiments. Indeed, the distance (R) between the 5′ end of miR-155 and the Trp214 of HSA within the HSA/miR-155 complex has been assessed by the means of FRET measurements, applying the donor lifetime variation method [36]. A solution of 5 µM HSA with miR-155-Atto390 (1:1) (volume/volume) has been prepared in order to register the fluorescence decay of the HSA/miR-155-Atto390 (DA) complex, where the co-presence of Donor (D) (Trp214) and Acceptor (A) (Atto390) enabled the long-range non-radiative energy transfer ($E_{FRET}$) in the excited state. The occurrence of such a process, in competition with the fluorescence deactivation pathway, is demonstrated by the clearly reduced fluorescence decay of HSA/miR-155-Atto390 (DA) in comparison to the decay of the HSA/miR-155 (D) complex (1:1) solution as observed in Figure 3. These data have been again successfully fitted by Equations (2) and (3), as witnessed by the weighted residuals shown at the bottom of Figure 3. The corresponding average fluorescence lifetime values, <τ_DA_> and <τ_D_>, have been inserted into Equation (5) for the evaluation of $E_{FRET}$, which, in turn, has been used to determine the distance, D_DA_, between A and D, through Equation (6) (see Section 3.3.3). We finally found D_DA_ = (3.9 ± 0.2) nm, a value very close to that obtained for the HSA/miR-4749 of (4.3 ± 0.5) nm. The determination of such a distance represents a useful starting point for the characterization of the binding site between HSA and miR-155.

### 2.4. Computational Docking 

Our experimental investigations by AFS and fluorescence techniques witness the formation of a specific complex between HSA and miR-155. Furthermore, FRET has led us to determine the distance between Trp214 and the dye (Atto390) bound to the 5′ end of miR-155 (D_DA_ = (3.9 ± 0.2) nm). Such structural information may be exploited to single out the possible interaction sites between the partners. In particular, it can be used to discriminate among the models of the HSA/miR-155 complex, as extracted by a computational docking between miR-155 and HSA. For miR-155, we have used the best model, shown in Figure 4A, as extracted by the procedure described in Section 3.4, while the X-Ray structure from the 1AO6 entry (from the protein data bank) [37] has been taken for HSA. The first 10 ranked models for the HSA/miR-155 complex, as derived by the docking, are collectively shown in Figure 4B. From a visual inspection, miR-155 may bind to HSA at three different sites, labelled as α, β and γ (see Figure 4B). We note that the α site of HSA is the same at which miR-4749 has been found to likely bind [24]. All these complexes have been preliminarily screened by evaluating the distance, D_DA_, between the center of the aromatic rings of the lateral chain of Trp214 of HSA and the 5′ end of miR-155, with the found values ranging in the 2.5–5.6 nm interval. By taking into consideration the experimental value of D_DA_ as well as the contribution of 0.1 nm due to the dye attached to the 5′ end of miR-155, we have selected those models for the complex whose D_DA_ distance is consistent with the corresponding experimental value. In particular, Model 3 and Model 5 (belonging to binding site α and Model 6 (belonging to binding site β) have been selected and submitted for further analysis; the corresponding distance of these complexes is reported in Table 1.

To investigate the stability of these models, and also to discriminate among them by evaluating the binding free energy, they have been submitted to a MD simulation, constituted by a preliminary equilibration of 3 ns, followed by a 10 ns long run for data collection (with five replicates for each model). The temporal evolution of the all-atoms RMSD for the various runs of the three models exhibits an initial increase, followed by an almost constant trend around values ranging between 0.35–0.45 nm. Such a behavior indicates the substantial stability of the formed complexes in the analyzed temporal window. Representative examples of D_DA_ during the 10 ns long trajectories for the three models are shown in Figure 5. We note fast fluctuations on which some jumps are superimposed; the latter indicating a local re-organization of the ligand at the binding site. Similar trends have been observed for the other runs. The average distances evaluated over all the MD runs for each model are reported in Table 1. In all cases, the distance does not significantly deviate from the initial value, consistently with the permanence of the ligand within the receptor site. 

The binding free energy, ΔG_B_ has then been evaluated from the MD runs by following the procedure provided in Section 3.6. The final value of ΔG_B_, together with various contributions, (ΔG_nonpol solv_, ΔE_MM_-TΔS, and ΔG_pol solv)_), is reported in Table 1. It should be remarked that the binding free energy values computationally evaluated commonly do not match those experimentally determined. Such a discrepancy has been attributed to several factors, including differences in the time window and sensitivity in the initial conditions (see ref. [38]). However, the computationally derived binding free energy is considered a valid tool for the screening of different structural models [39].

The nonpolar solvation term, ΔG_nonpol solv_, is negative, although rather small, in all three models. At the same time, the internal energy term, ΔE_MM_, is always negative, with the lowest value observed for Model 6. Furthermore, the entropic term, –TΔS_MM_, values are rather high, positive and similar in all the cases. Such a result indicates that the binding of miR-155 to HSA yields a slight reorganization of the molecules with an increase in entropy. In this respect, we mention that the structure of miRNAs is rather sensitive to the conditions of the external environment and that, in solution, they may sample different conformations [40]. 

Finally, the electrostatic term, ΔG_pol solv_, provides the most significant contribution to the binding free energy, and it is always negative and markedly different in the three models. These results indicate that electrostatics plays a central role in the formation of the complex, with a contribution from the entropic term. 

The resulting binding free energy is negative for all three models, and then all of them can be assumed to be energetically favorable, with Models 3 and 5 binding in almost the same HSA region with a similar topography. Remarkably, such a binding site coincides with that involved in the HSA/miR-4749 complex. Since Model 5 is characterized by a lower binding free energy than Model 3, we have submitted it to further analysis together with Model 6, in which, instead, miR-155 engages a different site of HSA. In particular, we closely addressed the role of the electrostatic forces in the complex formation by evaluating the electrostatic surface potential by the Adaptive Poisson–Boltzmann Solver (APBS) for these two models with a time step of 0.1 ns [41]. The final electrostatic surface potential for a representative structure of Models 5 and 6 for the complex, is shown in Figure 6 and Figure 7, respectively.

In both cases, the negatively charged miR-155 binds to a positively charged HSA pocket, with the two pockets being located in different regions of HSA. The electrostatic guide for the HSA/miR-155 complex finds correspondence with that found for the interaction between HSA and miR-4749. Notably, this matches the very recent observation that protein interactions with partners are largely dominated by electrostatics [42]. Although Model 6 is characterized by a lower binding free energy, both models could represent an appropriate description of the more likely complex structure. A graphical view of Model 5 and Model 6 is shown in Figure 8. In Model 5, the binding of miR-155 to HSA mainly involves domains I and III, however, at different subdomains. 

In summary, these results show that HSA can bind to miR-155 at two possible sites, characterized by positive charges. In both cases, the most significant contribution comes from the electrostatic term. Such a finding confirms what has previously been found for the interaction between HSA and miR-4749.

## 3. Materials and Methods

### 3.1. Materials

Single stranded RNA oligonucleotide with the sequence of human miR-155-5p (5′- uaa ugc uaa ucg uga uag ggg -3′), alone (miR-155, 6770 Da), labelled at the 5′ end with the ThiolC6 linker group (miR-155-ThiolC6, 7105 Da) and with the Atto390 fluorescent dye (miR-155-Atto390, 7281 Da), was purchased from Metabion (Planegg, Germany). The producer purified the oligonucleotides by HPLC-MS. They were resuspended in sterile TE (10 mM Tris-HCl, 1 mM Ethylenediaminetetraacetic acid (EDTA), pH 7.0) aqueous buffer solution and stored at 253 K in absence of light. Work surfaces and equipment were decontaminated using RNase*Zap*™ (Merck KGaA, Darmstadt, Germany). Prior to use, miR-155-ThiolC6 was incubated for 1 h with 100 mM Dithiothreitol (DTT) (Sigma-Aldrich Co.) in TE buffer at pH 8.0 in order to break the disulfide bond protecting the thiol (SH) moiety; the obtained miR-155-SH was eluited from a NAP10 column (GE Healthcare, Chicago, IL, USA) with Tris-HCl buffer (10 mM Tris-HCl, 150 mM KCl, pH 7.0) for the removal of DTT. The Human Serum Albumin (HSA) (66.5 kDa) was purchased from Sigma-Aldrich Co. (St. Louis, MO, USA) as a globulin free (purity degree > 99%) lyophilized powder and dissolved in the reaction medium, Tris-HCl buffer (10 mM Tris-HCl, 150 mM KCl, pH 7.0). Buffers were prepared using reagents from Sigma-Aldrich Co. and bidistilled water; after being microfiltered (Sartorius, Göttingen, Germany), they were stored at 277 K and thermalized at room temperature before use. 

### 3.2. AFS Experiments

Functionalization of tips and substrates used in AFS experiments was performed by following the procedures reported in refs. [11,32]). Briefly, the silicon nitride AFM tips (cantilever B, MSNL-10; Bruker Corporation) with a nominal spring constant, k_nom_, of 0.02 N/m were functionalized with miR-155 using a flexible linker, N-hydroxysuccinimide-polyethyleneglycol-maleimide (NHS-PEG-MAL, 3 kDa, hereafter PEG) (Iris Biotech, Marktredwitz, Germany). Such a linker was adopted in order to ensure miR-155 mobility and adequate distance from the inorganic surface, as well as to help discriminate specific unbinding events from unspecific ones through the characteristic non-linear trend of its stretching in force curves. The tips were cleaned in acetone (Sigma-Aldrich Co.) and ultraviolet (UV) irradiated for 30 min to expose hydroxyl groups. They were therefore incubated for 2 h at room temperature with a solution of 2% (volume/volume) 2-aminopropyl-triethoxysilane (APTES) (Acros Organics, Geel, Belgium) in chloroform (Sigma-Aldrich Co.), extensively washed with chloroform, and dried with nitrogen. The silanized tips were then immersed in a 1mM solution of PEG in dimethylsulfoxide (DMSO) (Sigma-Aldrich Co.) for 3 h, allowing the NHS-ester groups of the PEG to bind to the amino groups of APTES. After washing with DMSO and microfiltered bidistilled water, the tips were incubated overnight at 277 K with 10 μL of 10 μM miR-155-SH in TE buffer pH 7.0, enabling -MAL groups of the anchored PEG to react with the thiol moieties of miR-155-SH. Aldehyde-functionalized glass surfaces (PolyAn GmbH, Berlin, Germany) were incubated with 50 µL of HSA (10 µM) in Tris-HCl buffer pH 7.0 overnight at 277 K to promote covalent binding of proteins via their external lysine residues. Unreacted groups were passivated by incubation for 30 min with 1 M ethanolamine hydrochloride pH 8.5 (GE Healthcare, USA). All the samples were stored in the Tris-HCl buffer at 4 °C.

AFS measurements were performed at room temperature with the Nanoscope IIIa/Multimode AFM (Veeco Instruments, Plainview, NY, USA) in Tris-HCl buffer at pH 7.0. The force, F, was extracted by multiplying the cantilever deflection by its effective spring constant (k_eff_), is determined according to the procedure in ref. [43]. Force curves were collected by approaching the tip to different points of the substrate at a constant velocity of 50 nm/s while the retraction velocity was varied from 50 to 4200 nm/s. The spring constant of the entire system, k_syst_, is determined according to the procedure in ref. [44]. At each loading rate, more than thousands of force curves were acquired to guarantee information with statistical significance. Curves characterized by a nonlinear trend before the jump-off with the peculiar stretching features of the PEG linker were selected as specific unbinding events [29,45]. Ambiguous unbinding events were also analyzed by using the 1/f noise approach [46].

### 3.3. Fluorescence

All fluorimetric assays were conducted at 298 K with a FluoroMax^®^-4 Spectrofluorometer (Horiba Scientific, Jobin Yvon, Palaiseau, France), placing the analytes in a quartz cuvette with four optical faces and an optical path of 1 cm (Hellma, Munich, Germany). The samples were prepared by mixing stock solutions of miR-155 (c = 500, 100 and 20 µM in TE) with HSA (c = 10 µM in Tris-HCl) and diluting them with Tris-HCl buffer to obtain the desired concentration ratios; the final concentration of HSA was 5 µM in all the samples, while [miR-155] ranged from 1.5 µM to 50 µM, ranging from approximately 0.05·K_SV_^−1^ to 2·K_SV_^−1^ [47]. Measurements were repeated five times, and for each replicate, the sample solution was gently stirred and left to rest for 120 s, allowing the system to reach the equilibrium again. The mediated registered values, together with their standard deviations (SD), were used to calculate the fluorescence parameters (K_SV_, <τ>).

#### 3.3.1. Static Fluorescence

The fluorescence quenching experiments were carried out using the FluorEssence software (Horiba Scientific). Emission spectra were recorded in right-angle geometry as a signal/reference ratio, to be independent of the excitation intensity fluctuations of the Xe lamp; appropriate corrections for the instrumental response were applied. The spectral bandpass was set at 5 nm for both the excitation and emission monochromators. Under fixed excitation at 295 nm, emission profiles were collected from 305 to 580 nm, at increments of 1 nm and with an integration time of 0.50 s. Corrections for the Raman signal of the buffer and for the inner filter effect (for the solutions with A_295nm_ > 0.15) were performed. 

Fluorescence data were processed by means of linear fitting to the following Equation [35]:(2)$$\frac{F_{0}}{F}=1+k_{q}\tau_{q}\left[Q\right]=1+K_{\mathrm{SV}}\left[Q\right]$$
where F_0_ is the fluorescence intensity of the fluorophore (here HSA) in the absence of the quencher Q (here miR-155), F is the fluorescence intensity of HSA in the presence of miR-155, K_SV_ is the Stern–Volmer quenching constant, k_q_ is the bimolecular quenching constant, and τ_q_ is the lifetime of the fluorophore in the absence of a quenching agent. 

#### 3.3.2. Time-Resolved Fluorescence

Time-resolved fluorescence experiments were carried out using the DeltaHub^TM^ high throughput TCSPC controller (Horiba Scientific). The apparatus was set up to operate in reverse mode. The repetition rate (R.R.) of the excitation source, a pulsed nanoLED at λ = 295 nm (Horiba Scientific), was set at 1 MHz, and a TAC count rate of 10 kcps was adopted. The spectral bandpass was fixed at 5 nm for the excitation monochromator, while it was varied between 4 and 23.5 nm for the emission monochromator in order to mantain α = TAC count rate/R.R. = 1%. Scattered and emitted photons were detected at 346 nm, and such acquired fluorescence data was processed by the DAS6 software (Horiba Scientific) as a convolution of the impulse response function and the fluorescence decay through the Equation: (3)$$I\left(t\right)={\;a}_{0}+\overset{n}{\underset{i=1}{\sum }}a_{i}e^{\left(-\frac{t}{\tau_{i}}\right)}$$
in which I(t) is the time-dependent intensity, a_0_ accounts for the background, and a_i_ are preexponential factors representing fractional contributions to the time-resolved decay of the ith component with lifetime *τ*_i_. The goodness of the fit was judged in terms of both χ^2^ value and weighted fit residuals. The fluorescence lifetime, *τ*_i_, was then calculated by the following Equation:(4)$$\tau =\frac{\sum_{i=1}^{n}a_{i}\tau_{i}}{\sum_{i=1}^{n}a_{i}}$$
considering two exponential contributions.

#### 3.3.3. FRET

The dipole–dipole coupling requirement for FRET was met by adopting the Atto390 dye as the acceptor, since its absorbance overlaps the emission spectrum of the donor embedded in the studied system: Trp214 [11]. The E_FRET_ was evaluated by the donor lifetime variation method [36], according to [35]:(5)$$E_{\mathrm{FRET}}=1-\frac{<\tau_{\mathrm{DA}}>}{<\tau_{D}>}$$
where <τ_D_> is the average fluorescence lifetime of the donor when miR-155 is bound to HSA (1:1 ratio), while <τ_DA_> is its lifetime when the acceptor is attached to the 5′ end of miR55 in the HSA/miR-155-Atto390 complex. The distance D_DA_ between donor and acceptor was calculated from E_FRET_ through the relationship [35]:(6)$$E_{\mathrm{FRET}}=\frac{R_{0}^{6}}{\left(R_{0}^{6}+D_{\mathrm{DA}}^{6}\right)}$$
where R_0_ is the Förster radius, i.e., the donor–acceptor distance at which E_FRET_ equals 0.5 [48].

### 3.4. Modelling Procedures

The initial atomic coordinates of HSA were taken from the X-ray structure at 2.5 Å resolution (chain A of 1AO6 entry from the protein data bank) [37]. HSA is constituted by a single a-helix chain of 582 amino acids organized into three homolog domains (sites I, II and III). The structure of miR-155, not available, was obtained by the modelling procedure by following that developed in ref. [12]. Starting from the miR-155 sequence (reported in Section 3.1), the secondary structure, as well as the dot-bracket notation, were determined by RNAFOLD under default parameters [49]. The extracted information was used to derive the 3D structure using the RNACOMPOSER software [50]. The obtained best model was then submitted to a computational docking with HSA by HNADOCK [46]. The first 50 ranked models were first grouped by evaluating their structural differences in terms of the Root Mean Square Displacement (RMSD). Models differing among them for RMSD values of less than 0.1nm were grouped together; the first ranked model having been taken as representative of the corresponding group. Such a screening procedure allowed us to finally select ten models, named Model 1–10, and subject them to further refinements. All the figures for the biomolecules were created by Pymol [51] and VMD [52].

### 3.5. Molecular Dynamics (MD) Simulations

MD simulations of HSA, miR-155 and the HSA/miR-155 complex in water were carried out by the GROMACS 2018 package [53], using AMBER03 Force Field for the protein and miR-155 [54], and SPC/E for water [55]. All the molecular systems were centered in a cubic box of edge 9.0 nm^3^. Simulations were performed by following the procedures described in refs. [51,56]. Briefly, in each system, the box was filled with water molecules, to reach a minimum hydration level of 9 g water/g protein. The ionization states of protein residues were fixed at pH 7, and Cl^−^ or Na^+^ ions were added to keep the system electrically neutral. In particular, 34 Na^+^ were added to the HSA/miR-155 complexes, while 15 Na^+^ and 19 Na^+^ were added to HSA and to miR-155, respectively. H bonds were constrained with the LINCS algorithm [57]. The Particle Mesh Ewald (PME) method [58,59] was applied to calculate the electrostatic interactions with a lattice constant of 0.12 nm. Periodic boundary conditions in the NPT ensemble with T = 298 K and p = 1 bar with a time step of 1 fs were used. The other simulation conditions were the same as those used in ref. [24]. The trajectories were monitored by analyzing the RMSD, the Root Mean Square Fluctuation (RMSF), and the solvent accessible surface area (SASA) through the GROMACS package tools [53].

### 3.6. Calculation of the Binding Free Energy

The binding free energy, ΔG_B_, of the HSA/miR-155 complex was evaluated by the Molecular Mechanics Poisson–Boltzmann Surface Area (MM-PBSA) method, by following the same procedure as reported in refs. [58,59,60]. Briefly, the free energy, G, is expressed by: G = E_MM_ –TS_MM_ + G_solv_, where E_MM_ is the internal energy, TS_MM_ is the entropic term, and the G_solv_ the solvation contribution, decomposed into electrostatic (G_polar, solv_) and non-polar (G_nonpolar, solv_) parts [61]. The E_MM_ energy was evaluated from E_MM_ = E_elec_ + E_VdW_, where = E_elec_ is the protein–protein electrostatic and E_VdW_ is the Van der Waals interaction energy. The entropic contribution was estimated by the quasi-harmonic approach as reported in refs. [62]. G_polar, solv_ was evaluated by numerically solving the Poisson–Boltzmann equation with the Adaptive Poisson–Boltzmann Solver (APBS) software [41], with a 0.512 × 0.510 × 0.506 Å grid-spacing, and using the AMBER03 force field parameters, a probe radius of 1.4 Å for the dielectric boundary. The dielectric constant was set to 2 for the interior and to 78.5 for water [63]. The nonpolar part of the solvation contribution was obtained from G_non polar, solv_ = γ SASA + β, with γ = 2.27 kJ mol^−1^nm^−2^ and β = 3.84 kJ/mol [64]. Then, the binding free energy, ΔG_B_, of each model of the complex, was finally evaluated by: ΔG_B_ = Δ_complex_ − (ΔG_receptor_ + ΔG_ligand_), where each term was derived from the average of the values from 10 snapshots, recorded every 1 ps from the last 1 ns of the 10 ns long MD simulation runs, for each of the replicates.

## 4. Conclusions

AFS and fluorescence experiments allowed us to demonstrate the formation of a stable complex between HSA and miR-155 with an affinity constant K_A_ of 10^4^ M^−1^. Furthermore, time-resolved FRET measurements led us to determine the distance between the lone Trp214 of HSA, acting as a donor, and the acceptor chromophore labelling the miR-155 at its 5′ end. This structural information was exploited to discriminate among putative complex models as obtained by docking supported by computational modelling, molecular dynamics and binding free energy calculations. We highlighted, in HSA, two possible binding sites for miR-155. In both of them, the negatively charged miR-155 accommodates within a positively charged pocket of HSA; with this pointing out an essentially electrostatic drive, thus favoring the formation of the complex. Notably, one of the two binding sites on HSA is the same of that previously found for the interaction with miR-4749. Accordingly, it emerged that HSA is prone to bind negatively charged miRNAs, at specific sites. More generally, an active role for HSA as a carrier of miRNAs, aimed at a delivery or removal of miRNAs within the cell environment can be speculated about. Such a hypothesis finds correspondence with the intermediated affinity estimated for the complex which might legitimize a relatively stable binding of miR-155 followed by its easy release, assuring an adequate turnover. In summary, the possibility that HSA could regulate miRNA levels may offer new perspectives in the understanding of the mechanisms involved in the regulation of miRNAs.

## Figures and Tables

**Figure 1 ijms-23-10728-f001:**
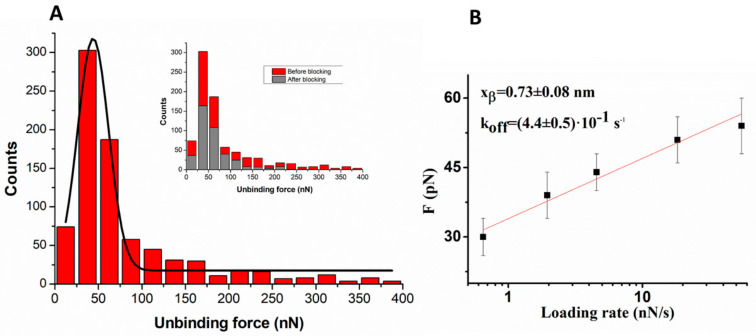
(**A**) Histogram of the unbinding forces for the HSA/miR-155 complex from AFS measurements carried out at a loading rate of 4.6 nN/s. The most probable unbinding force value (F*) has been determined from the maximum of the main peak of the histogram by fitting with a Gaussian function (black curve). Inset: Histogram of the unbinding forces for the HSA/miR-155 complex before and after blocking, at the same loading rate. (**B**) Plot of the most probable unbinding force, F*, vs. the logarithm of the loading rate for the HSA/miR-155 complex. The red continuous line is the best fit by the Bell–Evans model with Equation (1); the extracted values for the k_off_ and x_β_ parameters are reported.

**Figure 2 ijms-23-10728-f002:**
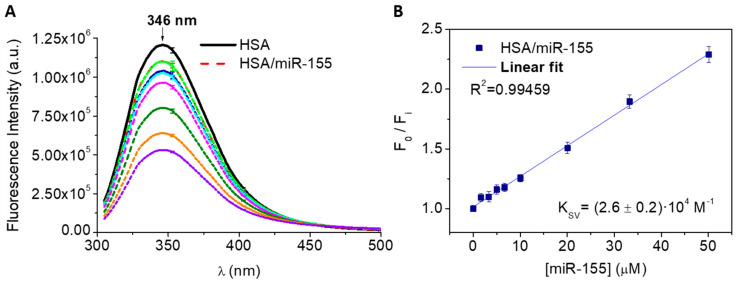
(**A**) Fluorescence quenching of 5 µM HSA in Tris-HCl buffer, pH 7.0, upon titration with miR-155 (1.5–50 µM); emission spectra have been collected using λ_exp_ = 295 nm, at 298 K. (**B**) linear fitting of the fluorescence intensity as a function of miR-155 concentration according to Equation (2); the extracted K_SV_ value being reported. The arrow indicates the peak of the spectra, while the bars indicate the errors.

**Figure 3 ijms-23-10728-f003:**
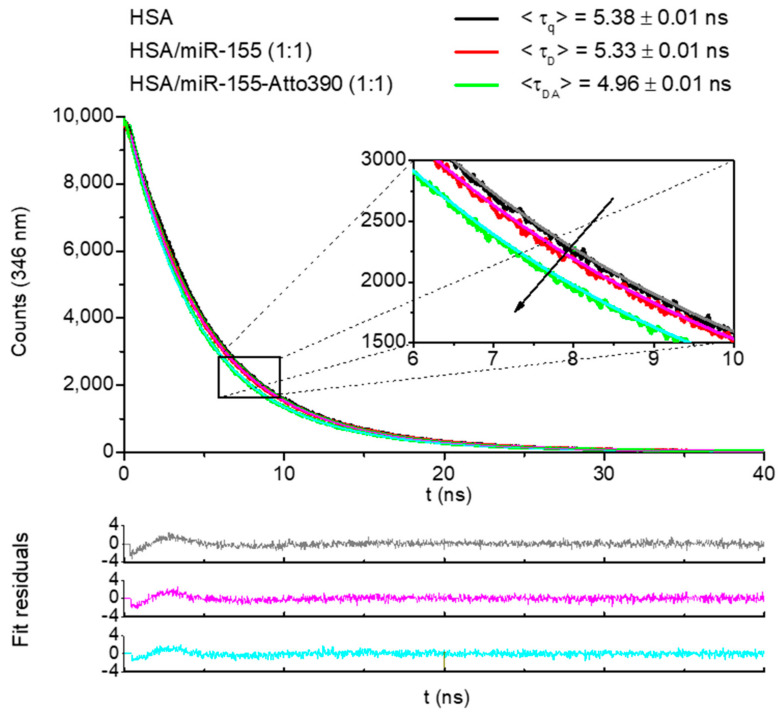
(**Top**): Time-resolved fluorescence decays of HSA, of the D (Donor) complex HSA/miR-155 (1:1), and of the DA (Donor-Acceptor) complex HSA/miR-155–Atto390 (1:1), acquired using λexc = 295 nm and λem = 346 nm, at 298 K; c = 5 µM in Tris-HCl buffer, pH 7.0. The respective average fluorescence lifetimes (<τ>), extracted by fitting the decays according to Equations (3) and (4), are reported. (**Bottom**): weighted fit residuals.

**Figure 4 ijms-23-10728-f004:**
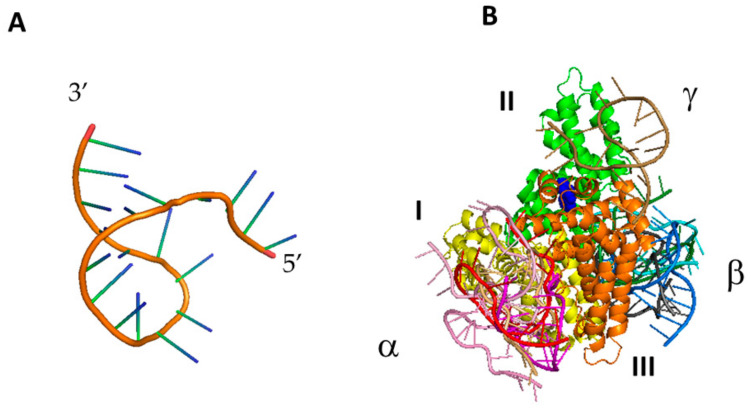
Graphical representation of: (**A**) model for miR-155. (**B**) Collective representation of the ten best models for the HSA/miR-155 complex. Regarding HSA, domain I is colored in yellow, domain II in green and domain III in orange; Trp214 being shown as blue spheres. The three possible binding sites of HSA for miR-155 are indicated by α, β and γ.

**Figure 5 ijms-23-10728-f005:**
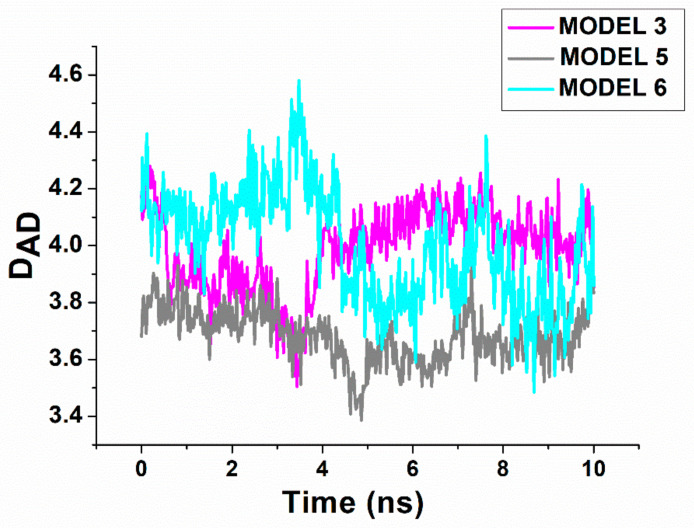
Temporal evolution of the D_DA_ distance between the 5′ end of miR-155 and the aromatic ring center of the lateral chain of Trp214, for Model 3 and Model 5 of the complex during a MD run.

**Figure 6 ijms-23-10728-f006:**
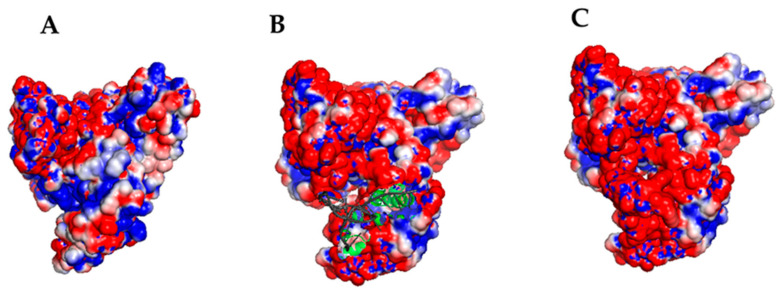
Electrostatic surface potential visualization, as evaluated by APBS for the structure of Model 5 for the HSA/miR-155 complex at the end of a 10 ns long run. (**A**) HSA without showing miR-155; (**B**) HSA with miR-155, represented by a cartoon (grey); (**C**) HSA/miR-155 complex with miR-155, represented by spheres. Red spheres indicate negative charges, while blue spheres represent positive charges.

**Figure 7 ijms-23-10728-f007:**
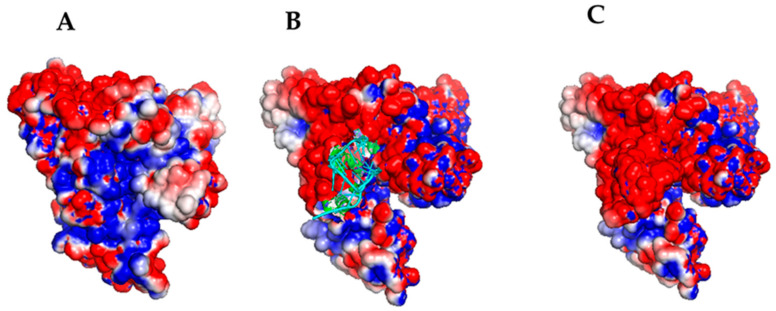
Electrostatic surface potential visualization, as evaluated by APBS for the structure of Model 6 for the HSA/miR-155 complex at the end of a 10 ns long run. (**A**) HSA without showing miR-155; (**B**) HSA with miR-155, represented by a cartoon (cyan); (**C**) HSA/miR-155 complex with miR-155 represented by spheres. Red spheres indicate negative charges, while blue spheres indicate positive charges.

**Figure 8 ijms-23-10728-f008:**
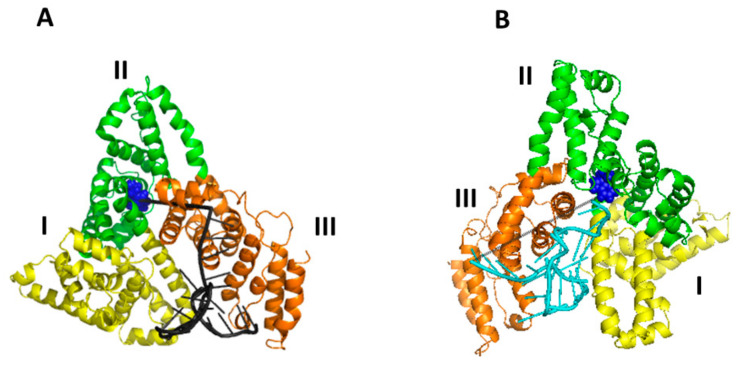
Graphical representation of the two proposed best models for the complex between HSA and miR-155: (**A**) Model 5 with miR-155 represented in grey; and (**B**) Model 6, with miR-155 represented in cyan. In HSA, domain I is colored in yellow, domain II in green and domain III in orange; Trp214 is shown as blue spheres. The distance D_DA_ between Trp146 and the 5′ end of miR-155 and the center of the aromatic rings of the lateral chain of Trp214 of HSA is marked by gray dashed lines.

**Table 1 ijms-23-10728-t001:** Some properties of the three best models for the HSA/miR-155 complex. D_DA in_ is the initial distance between the 5′ end of miR-155 and the aromatic ring center of the lateral chain of Trp214, D_DA ave_ is the average over the MD simulations. ΔG_B_ is the binding free energy calculated from ΔG_B_
*=* Δ_complex_ − (ΔG_receptor_ + ΔG_ligand_), from the various terms: ΔG_nonpol solv_ is the nonpolar contribution to the solvation term, ΔE_MM_, the internal energy, −TΔS_MM_, the entropic term, and finally the ΔG_pol solv_ is the electrostatic contribution to the solvation term.

MODEL #	D_DA in_ (nm)	D_DA ave_ (nm)	ΔG_nonpol solv_ (kJ/mol)	ΔE_MM_ (kJ/mol)	−TΔS (kJ/mol)	ΔG_pol solv_ (kJ/mol)	ΔG_B_ (kJ/mol)
**Model 3**	3.8	3.8 ±0.1	−31	−175	872	−1300	−634
**Model 5**	4.0	3.7 ± 0.1	−30	−198	878	−1600	−950
**Model 6**	4.2	4.0	−10	−392	919	−2400	−1883

## Data Availability

Not applicable.

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
