# Peer review of "Interaction of miR-155 with Human Serum Albumin: An Atomic Force Spectroscopy, Fluorescence, FRET, and Computational Modelling Evidence"

_ijms, 2022, doi:10.3390/ijms231810728_

Round 1

Reviewer 1 Report

This paper studies the interaction between Human Serum Albumin (HSA) and microRNA(miRNA) 155 through computational methods. The content is similar to the authors’ other paper that investigated the interaction between HSA and miR-4749. According to the result, they concluded that native HSA is a suitable miRNA carrier under physiological conditions for delivering to appropriate targets. I have serious concerns about this paper.

Specific comments.

1.      Circulating miRNAs have been used as many disease biomarkers. It is known that serum can carrier miRNAs. This manuscript concludes that native HSA is a suitable miRNA carrier. This does not seem to be an important result, as HSA may carry more miRNAs than just miR-4749 and miR-155.

2.      The affinity constant and distance were calculated. But what is the use of these scores in understanding the function of miRNA in pathologies?

3.      There should be a hyphen between the letters and the numbers of the miRNA notation.   All notations of miRNAs need to be revised, e.g. miR155 →”miR-155.

4.      Figure 1. (A). You fit the count with a Gaussian model. But the count skews to left. It is not a symmetrical shape. It is not suitable to fit with a normal distribution. Other distributions such as gamma distribution might be suitable.

Author Response

  1. Circulating miRNAs have been used as many disease biomarkers. It is known that serum can carrier miRNAs. This manuscript concludes that native HSA is a suitable miRNA carrier. This does not seem to be an important result, as HSA may carry more miRNAs than just miR-4749 and miR-155.
  2. The affinity constant and distance were calculated. But what is the use of these scores in understanding the function of miRNA in pathologies?

We believe that a check on the effective capability of HSA to specifically bind miR-155, which is very importantly related with several cancer pathologies, could be of some interest. In particular, the information on the topology/structure of the binding site between HSA and this microRNA, which has been extracted by spectroscopic techniques (and not only computational ones) might be used to design more appropriate carriers, or to infer more details on the strategy adopted by this microRNA to bind to its physiological receptors. More generally, the interaction of HSA with miRNAs (or even circRNAs), not fully elucidated yet, could help to clarify the interaction network of these oligonucleotides.

  1. There should be a hyphen between the letters and the numbers of the miRNA notation.   All notations of miRNAs need to be revised, e.g. “miR155” →”miR-155”.

The text has been corrected according to the Reviewer’s indications. Along the same direction Figures 2 and 3, both featuring the previous notation, have been modified.

  1. Figure 1. (A). You fit the count with a Gaussian model. But the count skews to left. It is not a symmetrical shape. It is not suitable to fit with a normal distribution. Other distributions such as gamma distribution might be suitable.

Indeed, the histograms of unbinding forces are expected to be slightly asymmetric and skewed towards low force values (see  the newly added ref.[27] in the manuscript).  However, it has been verified that to extract the most probable unbinding force, a fit by a Gaussian distribution yields the same results (see ref.[25] in the manuscript).  Such an aspect has been now explicitly mentioned in the manuscript (Pag.3 Lines 136-141).

Reviewer 2 Report

In this work interaction between Human Serum Albumin (HSA) and microRNA 155 (miR155) have been investigated with several different experimental and a computational methods. The work is interesting, carefully prepared and earns probably high interest justifying publication in the International Journal of Molecular Sciences. The references are upto date, the structure of the manuscript is appropriate, the data collected by the techniques applied support the conclusions. I suggest publication with minor revision considering the following remarks:

The interaction free energy estimated around 1000 kJ/mol (depending on the model applied: -634 kJ/mol, -950 kJ/mol, -1883 kJ/mol) while the complex stability constant (log K) estimated around 4. The stability constant suggest only a few tens kJ/mol Gibbs free energy changes during the formation of complexes. Maybe I misunderstood something, but I suggest to check the energy values and also why the entropy decreases such much (authors highlights that the –TdeltaS is always positive). If this is the case, which interaction cover the binding ?

Figure 7. The electrostatic potential map prepared under 10 ns long run. What was the resolution (time step) in time during this calculations ?

Author Response

  1. The interaction free energy estimated around 1000 kJ/mol (depending on the model applied: -634 kJ/mol, -950 kJ/mol, -1883 kJ/mol) while the complex stability constant (log K) estimated around 4. The stability constant suggest only a few tens kJ/mol Gibbs free energy changes during the formation of complexes. Maybe I misunderstood something, but I suggest to check the energy values and also why the entropy decreases such much (authors highlights that the –TdeltaS is always positive). If this is the case, which interaction cover the binding ?

Indeed, the binding free energy values of biomolecular complexes as evaluated by MD simulations generally deviate from those experimentally determined. The lack of reproducibility stems primarily from the chaotic nature of classical MD simulations, inadequacy of the force field parametrization, sensitivity of trajectories to their initial conditions, etc. (see the newly added ref.[38] in the manuscript). On the other hand, simulated binding free energy can more reliably be used for comparison within a well defined framework, as it has been done in our case in which free energy has been sued to compare different complexes as obtained by docking. Such an aspect has been now mentioned in the text (Pag.9 Lines 343-348).

Concerning the entropy, our interpretation is that the binding between HSA and miR-155 promotes a conformational change of the biomolecular system, and in particular of the miR-155 which is prone to undergo structural changes; such an aspect having been now mentioned in the text together with the newly added ref.[63] (Pag.9 lines 361-363).

  1. Figure 7. The electrostatic potential map prepared under 10 ns long run. What was the resolution (time step) in time during this calculations ?

The time step used in the electrostatic potential is of 0.1 ns; such information has been now added to the manuscript (Pag.9 Line 376).

Reviewer 3 Report

Some minor checks are required before being published in IJMS.

Author Response

  1. R83-R91 - This paragraph expresses a conclusion, therefore it does not belong in the Introduction section, but rather in the discussion chapter or even in the Conclusions section.

The last paragraph containing a sentence with conclusive remarks has been removed from the Introduction. This part has been only mentioned in the Conclusions.

  1. R158 – The affinity constants KA must be reported in M-1.

Corrected accordingly.

  1. R 202, 203 -the lifetime values are not written correctly – see R 207 for kq

Corrected accordingly.

  1. The authors cited themselves in a proportion of over 25%. If possible, please find other bibliographic sources that support the ideas presented.

Although all the citations of our previous papers have been appropriately used to describe a specific part of the work, to address the Reviewer’s suggestion, some of them have been substituted with papers of other authors or removed.

Round 2

Reviewer 1 Report

I suggest accepting this paper.